# Coat Color in Local Goats: Influence on Environmental Adaptation and Productivity, and Use as a Selection Criterion

**DOI:** 10.3390/biology12070929

**Published:** 2023-06-29

**Authors:** Pablo Arenas-Báez, Glafiro Torres-Hernández, Gabriela Castillo-Hernández, Martha Hernández-Rodríguez, Ricardo Alonso Sánchez-Gutiérrez, Samuel Vargas-López, Juan González-Maldonado, Pablo Alfredo Domínguez-Martínez, Lorenzo Danilo Granados-Rivera, Jorge Alonso Maldonado-Jáquez

**Affiliations:** 1Unidad Regional Universitaria de Zonas Áridas, Universidad Autónoma Chapingo, Bermejillo, Durango 35230, Mexico; parenasb@chapingo.mx; 2Colegio de Postgraduados, Campus Montecillo, Montecillo, Texcoco 56264, Mexico; glatohe@colpos.mx (G.T.-H.); castillo.gabriela@colpos.mx (G.C.-H.); hernandez.martha@colpos.mx (M.H.-R.); dominguez.pablo@inifap.gob.mx (P.A.D.-M.); 3Facultad de Estudios Superiores Cuautitlán, Universidad Nacional Autónoma de México, Cuautitlán Izcalli 54714, Mexico; 4Instituto Nacional de Investigaciones Forestales Agrícolas y Pecuarias, Campo Experimental Zacatecas, Calera, Zacatecas 98500, Mexico; sanchez.ricardo@inifap.gob.mx; 5Colegio de Postgraduados, Campus Puebla, Cholula 72760, Mexico; 6Instituto de Ciencias Agrícolas, Universidad Autónoma de Baja California, Mexicali 21750, Mexico; juan.gonzalez.maldonado@uabc.edu.mx; 7Instituto Nacional de Investigaciones Forestales, Agrícolas y Pecuarias, Campo Experimental Valle del Guadiana, Durango 34170, Mexico; 8Instituto Nacional de Investigaciones Forestales, Agrícolas y Pecuarias, Campo Experimental Genera Terán, General Terán 67400, Mexico; 9Instituto Nacional de Investigaciones Forestales, Agrícolas y Pecuarias, Campo Experimental La Laguna, Matamoros 27440, Mexico

**Keywords:** genetic improvement, productivity, small farmers

## Abstract

**Simple Summary:**

The physical characteristics that are observed at first sight in an animal (for example, coat color) tell us a lot about its capacity to adapt to the surrounding environment. In this review, we address the issue of color in goats and its direct effect on meat and milk production to design a tool for small producers (mainly in rural areas) that will allow them to have selection criteria that are very easy to apply.

**Abstract:**

This paper aims to review, systematically synthesize, and analyze fragmented information about the importance of coat color in local goats and its relationship with productivity and other important traits. Topics on current research on color expression are addressed, the relationship that has as a mechanism of environmental adaptation, its relationship with the production of meat, milk, and derivates, and the economic value of this characteristic. The use of this attribute as a tool to establish selection criteria in breeding programs based on results reported in the scientific literature is significant, particularly for low-income production systems, where the implementation of classic genetic improvement schemes is limited due to the lack of productive information, which is distinctive of extensive marginal or low scaled production systems around the world.

## 1. Introduction

Small ruminants play an essential role in the economy of millions of people worldwide, particularly for those in rural communities with a high margination rate, where people have learned to breed these animals under several environmental conditions and where goats are one of the most successful species [1]. Due, in principle, to their great capacity for adaptation and higher efficiency in consumption and utilization of fodder, which has allowed them to develop, produce and reproduce in harsh environments, they are commonly known as a “poor man’s cow” [2,3]. In addition, goats are more resilient to higher temperatures, possess a small corporal size, and have mobile lips, which allow them to choose the most nutritious parts of the plants and proportionally produce less metabolic heat than cattle and sheep [4]. 

These characteristics allow goats to hold a high production potential and make them an ideal species, key to providing animal protein in the face of two scenarios of global concern: global warming and an increase in meat demand due to the rising human population [5]. Nevertheless, the low investment capacity of the producers, developed under extensive grazing, inhibits the adoption of technologies, which limits the productive capacity of livestock; therefore, it is necessary to generate low-cost and easy-adoption technologies according to the production system requirements [3,6]. 

In general, goats exploited in extensive systems are native (local) animals, characterized for having a heterogeneous phenotype, where skin and coat coloration show a tremendous variation, which represents a precious genetic reservoir [7]. Implementing indiscriminating crossing schemes without order or production-defined objectives has put serious extinction risk to these populations [8], limiting the possibility of generating improved productive schemes to help face the current environmental and market challenges [9]. Because of this, adapting selection schemes and classic improvement to small herds is one of the main issues of genetic improvement in local breeds [10].

Some alternatives have focused on providing the producer with practical tools to select replacements based on appearance [11], considering coat color as a selection criterion for these local populations, as it is an easy-to-recognize feature that, moreover, has been widely used as a phenotypical selection model [12] and as a feature of local identification [13]. However, the related information on this topic is limited [14], particularly for local goats.

Based on the above, the objective of this review is to synthesize and analyze the available information on the coat color of native goats, focusing on the color expression, adaptation to the environment, its relationship with general productivity, economic importance, and incorporation of this characteristic into selection schemes, as part of genetic improvement programs in marginal or small-scale systems.

## 2. Expression of Coat Color and Its Role in Racial Conformation

Coat pattern in mammals is determined by the distribution of two types of melanin, eumelanin (black/brown pigments) and pheomelanin (red/yellow pigments), for which production and relative amounts of pigment in melanocytes are controlled by the Agouti gene [15]. In goats, the counterpart of this gene is the one of the Agouti Signaling Protein (ASP), which demonstrates a high expression in white coat specimens [16] because it favors the synthesis of pheomelanin by blocking the union of the α-melanocyte-stimulating hormone (α-MSH) with the melanocortin 1 receptor (MC1R) [17], limiting with it the synthesis of the necessary enzymes for the production of eumelanin. 

In this sense, coat coloration in mammals is determined by the amount, quality, and distribution of melanin in the body, which is a complex cellular process that includes differentiation and maturation of the melanocytes. In this, a complex regulatory net takes place, formed by the interaction of various genes and by the presence of different epigenetic states [18], starting with the signalization induced by MC1R and finishing with the regulation of Tyrosinase (TYR), Tyrosine Protein Related 1 (TRP1), and TRP2, which are required for melanin synthesis [19]. Although, in goats, the reports of the association between coat color, polymorphisms, and gene expression are just coming to light [13,16,20], the mechanisms by which these genes act to produce color in the coat are not well defined [21].

Coloration in animals can be a powerful model for studying the genetic and epigenetic mechanisms that determine the phenotype [22] and the transgenerational heritage of epigenetic marks [18]. Therefore, coat color has been considered the main feature for developing new breeds, eventually generating closed populations [23]. In fact, processes of domestication and dispersal of goats induced changes in numerous morphologic features, driven by targeted selection, genetic drift, isolation, and founder effect, giving origin to a wide range of shapes and an extraordinary color range [24]. For example, the brown goats of the Sichuan basin in China and the northeast goats from Brazil became very distinctive due to geographic isolation [7,25]. 

In the same way, artificial selection has triggered the involuntary fusion of characteristics related to the domestication syndrome (coat color and droopy ears), as these characteristics have reduced importance in domesticated animals, limiting the evolution of camouflage coloration to avoid predators and sexual dimorphism, as shown in horses and pigs [26]. The aforementioned was realized without considering that the composition of the coat coloration is a complex process that is dictated by the content of pigments and minerals in the fur; notwithstanding, the molecular base of the heritage for this feature has been poorly understood in goats, since there are many candidate genes involved in pigmentation that have not been sequenced yet (or their variability levels have not been quantified) [27,28]. 

In addition, extensive searches into goat germplasm through genome-wide association studies (GWAS) or restriction fragment length polymorphism (RFLP), associating candidate genes and coat color, have expanded the catalog of color genetic variants (Table 1). In cases such as the breeds in Switzerland, Pakistan, and Africa, molecular signatures have been determined for key genes in melanogenesis [13] that, along with the comparison of transcriptomes, have contributed to understanding the color expression in goats [16]. Furthermore, while it has been found that the Agouti gene exerts a critical effect on the synthesis of pigmentation [12], this effects depends on the methylation grade on Cytosine Phospho Guanine (CpG) retrotransposon intracisternal A particle (IAP) [29]. However, some techniques, such as quantitative real time-polymerase chain reaction (qRT-PCR) have shown that there is no single locus explaining color divergence, and the conclusion is that this is a multilocus answer [30] with prominent pleiotropic, epistatic, and non-epistatic effects influenced by the diet, age, and environment [31]. 

On the other hand, through techniques such as RT-PCR and Single Nucleotide Polymorphism (SNP), approximately 80 genes that modify coat color in goats have been identified, and half of the functional proteins codified by these genes are expressed in the melanosome [28]. Hence, it is possible to explore the relationship between the genetic changes in the populations and the phenotypic variation in terms of genomic indicators, identifying variants that can be part of the known genes [32]. In this sense, the research that associates the genotype–phenotype found 16 Mendelian traits and 10 causal mutations in goats, which is lower than those detected in cattle and sheep [24]. Nevertheless, a consistent pattern has not been found by which the light-colored goat breeds show a higher number of copies of the ASIP gene than observed in the dark-colored breeds. Even so, there is evidence that suggests that the duplication of the ASIP gene can be associated with the light color in some breeds, whereas ASIP codifies a protein that binds to MC1R and stimulates the synthesis of pheomelanin instead of eumelanin, as noted before [27,33]. Likewise, Becker et al. [34] found evidence that the brown coat in copper neck goats could be dominant but suggest continuing with the functional experimentation or selective breeding to verify the results, as other findings indicate that the brown/black coloration functionally depends on the genotype MC1R [35].

Based on the abovementioned and considering the advances made in genetic engineering, the idea of “editing” the coat pattern in goats has surfaced as an alternative for developing more resistant specimens which are adapted to adverse agro-environments for animal production [36]. In this regard, Laible et al. [37] verified the causal nature of the mutation of the PEML gene to dilute the color of the dark coat in cattle. Van Buren et al. [38] identified a variant of the MLPH gene that causes the dilution of the color in dogs. Wang et al. [39] found that one edition of 1bp in the TBX3 gene decreases the expression in the deposition of the pigment of the cape in donkeys. Zhang et al. [40] manipulated the gen ASIP in sheep via CRISPR-Cas9, resulting in a wide variety of color patterns.

With these findings, genetic editing in goats will likely become feasible. Thus, Wu et al. [41] and Li et al. [36] found that the single-stranded endogenous microRNAs miR-129-5p and miR-27a affect the formation of color by decreasing the expression of various genes, among which, tyrosinase (TYR) stands out, which can be used as a suppressor in the formation of the coat color in goats. Therefore, the genetic analysis based on the variation in the number of copies of genes and in the post-transcriptional silencing can help to understand the genomic architecture in a better way, and deeper knowledge offers the potential to help breeders to design more efficient selection strategies for genetic improvement [42]. Martin et al. [43] pointed out that genomic information can help to detect signatures for undesirable colors, as in the case of the pink color in Saanen goats. 

Other findings in mice explained the epigenetic mechanism, whereby the repressed expression of IAP inside the Agouti gene is maintained. The silencing of transcriptional activity is related to methylation of CpG sites at the DNA level, as well as to trimethylation of lysine 27 of histone 3 (H3K273me3) and trimethylation of lysine 20 of histone 4 (H4K20me3); therefore, repression of Agouti gene expression has been found to occur at the DNA levels of methylation and of post-translational modification of histones [44]. There are reports on IAP having six CpG sites in the long 3´-end terminal repetition (LTR) where methylation happens, which is the same one that is accompanied by the repressive mark H4K20me3 [45]. Both epigenetic modifications produce a repressive genetic state, and the individuals are phenotypically darker (methylated), contrasting with the Agouti yellow coloration (unmethylated) derived from the hypomethylation of IAP, i.e., the marks of deacetylation in H3K and H4K. On the other hand, the unmethylated CpG marks and histone acetylation are associated with the formation of euchromatin, a state that allows the transcription of the mutant Agouti, resulting in the systemic production of a protein that, under normal conditions, is expressed in the skin [46,47]. In this respect, this functional variation has been studied by using methyl as a dietary supplement; thus, the nutritional possibilities of epigenomic modulation have been expanded, particularly in developing environments, which could facilitate the manipulation of the coat color in goats through dietary manipulation.

**Table 1 biology-12-00929-t001:** Genetic diversity described for color codification in local goats.

Breed	Candidate Gene or Locus	Associated Color	Reference
Liaoning Cashmere	MC1R	Red/Black/White	[30]
	TYRP1	Brown	[30]
	KIT (exón 13)	White/Black	[28]
Boer	KITLG	White	[20]
Nanjiang	RALY	Brown/Black	[20]
	RALY-ELF2S2-ASIP	Yellow	[20]
Black Macheng	Agouti y AHCY	Black	[20]
	MC1R	Black	[48]
Mongolian Black Dazu Cashmere	miR-129-5p	Black	[36]
Black Yunnan	ASIP	Black	[49]
Meadow brown	ASIP	Brown	[49]
Iranian Markhoz	ASIP, ITCH, AHCY y RALY	Black/Brown	[50]
Fossés, Poitevine, Provencale, Pyréneés	ADAMTS20	Black/Brown/Gray	[51]

MC1R (Melanocortin 1 Receptor); TYRP1 (Tyrosinase Related Protein 1); KITLG (KIT Ligand); RALY (RALY Heterogeneous Nuclear Ribonucleoprotein); ELF2S2 (E74 Like ETS Transcription Factor 2); ASIP (Agouti Signaling Protein); AHCY (Adenosylhomocysteinase); KIT (KIT Proto-Oncogene, Receptor Tyrosine Kinase); ITCH (Itchy E3 Ubiquitin Protein Ligase); ADAMTS20 (ADAM Metallopeptidase With Thrombospondin Type 1 Motif 20).

## 3. Environmental Adaptation

The characteristics of the external surface of an animal’s body (skin and coat) are significant for their relationship with the environment, as this is the first protection barrier, and it can differ in color, depth, and length [1]. Coat color is a genetic factor of adaptation to different climates [52] because the thermoregulatory response is different between colors, with superiority noted in breeds with different shadows in contrast to animals with light or white coating, which reflect between 50 and 60% of direct solar radiation [53,54]. 

The preceding information confirms that dark coats absorb more heat than lighter coats. Nevertheless, penetration into the skin depends on the structure of the coat and its color [55]. Because of this, Castro-Lima et al. [56] discussed the capacity of short fur and dark skin to provide higher protection against solar radiation compared to short fur and light skin. Some reports indicate that coat color affects every heat tolerance criterion (rectal temperature, respiratory rate, cardiac frequency caloric stress index, and some hematologic parameters such as aldosterone concentration), suggesting that goats with different coat colors possess different thermoregulatory abilities [57,58]. Therefore, the tolerance and adaptation criterion of the animals is determined by a heat transfer to the skin nucleus through the arteriole’s dilatation to the extremities, ears, and muzzle, allowing an increase in peripheral blood flow and facilitating heat transfer [59].

Evidence of the above has been reported [58,60,61] when observing a lower urinary frequency in dark-coated animals, while those who are light-coated have a higher solar radiation reflection, lower heat absorption, and lower surface temperature, without differences in the degree of evapotranspiration between animals with dark and light coats, so this parameter must be related to other characteristics such as fur morphology and skin color [62]. Other researchers [63] point out that black goats are better at adapting to cold, as the black pigment helps them warm up faster. Similarly, Ferreira et al. [54] state that, in rainfall seasons, the black color lets the goats catch more sun energy in cold mornings, supporting homeostasis. 

In this regard, although the literature suggests that light-colored goats have a more significant advantage over those with dark colors concerning their thermoregulatory mechanisms in warm environments, this information is mostly valid for environments where the temperature does not vary much throughout the year. However, in arid regions, extreme temperatures are present during the day and within the seasons of the year; in order to select breeds or individuals well adapted to these environments, it seems necessary to carry out more extensive studies that include, in addition to coat color, skin color and texture, size, and fur thickness, as well as its distribution and shape in the body.

Since black-colored goats dominate desert areas, this may be because they have the advantage of facing direct exposure to solar radiation and can consume water equivalent to 35% of their body weight in just one drink; furthermore, some local genotypes (Black Bedouin) can be limited in water consumption for up to 4 days, which helps them to adapt effectively to the environment. Furthermore, producers in Nigeria do not select animals with white coats because they believe they are more susceptible to attack by predators and thefts, having a higher risk when goats wander away from the herd [3,64].

On the other side, some differences in fur color, generally, are attributed to genetic and non-genetic factors, including the age or intensity of solar radiation [65]. Because of this, and given the controversy found in the current literature, it is not possible to establish a coat color as the ideal for a given climate or environment, so further research is necessary to evaluate, in a particular way, the productive response of the animals with different coat colors in every environment.

## 4. Coat Color and Productivity in Local Goats

The productivity of goats and its relationship with coat color is perhaps one of the least-studied topics because color is a qualitative characteristic that cannot be measured based on a scale [66]. Nevertheless, there is the knowledge that coat color is inheritable, and some authors suggest that it is not clear if the selection based on this criterion could benefit productivity [67]. Shoyombo et al. [68] concluded that, even if the coat color seems weak, it has a moderate influence on the morpho-structural composition of the animals. Likewise, Choudhury et al. [69] indicate that this characteristic in Black Bengal goats affects the morphometry and production; this has led to the study of the possibility that the coat color is associated with reproductive characteristics and some productive features [70].

Evidence of the preceding is that there is a greater preference for brown or reddish male goats with black or brown spots or stripes, as they reproduce more (and more easily) than other colored males; in addition, these have a higher market value, while males with a dominant or entirely black coat are less desirable due to their low economic value [64]. Even though the black color protects against photochemical damage, in addition to the fur, the dark skin, with an abundance of eumelanin, helps to protect the epidermis from UV rays [54], which helps to improve reproductive parameters [71].

Choudhury et al. [69] observed that color affects the kidding interval and the litter size in Black Bengal goats. In these goats, solid colors had the shortest kidding interval among brown goats. Daramola & Adeloye [72] reported that the color interferes with prolificacy, fecundity, and litter size at birth and weaning, as well as in weights at weaning in West African Dwarf goats, and Baenyi et al. [73] reported that heat stress mainly affects the females more than the black-colored males.

Black Bengalese goats performed better in terms of birth weight, age at puberty, daily weight gain, milk production, lactation length, age at first birth, and litter size than goats of other colors [74]. Nevertheless, the effect of coat color on the productivity of goats is controversial, since some studies indicate that coat color has no relationship with productive performance [75]. Therefore, it is a subject that still requires more profound studies, similar to those carried out in mice, where researchers demonstrated that the expression of the coat color by dominant mutations in the Agouti gene and pre-existing epigenetic states are associated with weight gain and fatty acid synthesis by the insulin and Agouti response elements in the FASN gene (Fatty Acid Synthase) [76,77].

## 5. Milk and Meat Production

The evidence found regarding milk and meat production suggests that coat color is directly related to the productive efficiency of the animals (Table 2), since the coat coloration influences the caloric load that, when significantly increased, leads to a decrease in the rate of metabolic heat production, causing adverse effects for productivity [74]. For example, during lactation, the water amount in the organism decreases, causing a decrease in milk production and ejection, and affecting the synthesis of the growth hormone and prolactin [59]. In the same way, an increase in milk production relates to a decrease in the intensity of the pigmentation in the coat coloration [72]. For their part, Gupta and Mondal [78], in their review, infer that the loss in production occurs in combination with the advance in lactation and thermal stress, which induces a drastic reduction in the quantity and quality of the produced milk.

On the other hand, according to their culture, some African villages believe that breeding or eating some specific-colored animal has higher religious and health benefits, although this has not been confirmed [79]. Black goats from the Salem breed have better meat quality than Osmanabadi goats, which has links with higher resilience to thermal stress in black goats [80]. This information can be helpful, as these results broaden the vision of identifying agroecological zones for specific breeds, which will keep and increase livestock production in harsh environments.

Regarding growth rate, weight gain, and survival rate, animals with black coats stand out compared to light-coated specimens [72]. Hossain et al. [81] found that Dutch Belt goats showed significantly higher weights and growth rates between 3 and 6 months old, compared to other fur color variants. Other results indicate that brown-and-white patterned goats were heavier than other colored goats [69]. The opposite case is the one reported by Mabrouk et al. [82], that, in local Tunisian goats, there was no effect of coat color on body weight.

While there are contradictory results at the productivity level in the literature, this characteristic must undergo careful analysis in order to avoid adverse effects in local populations, as it has happened in other species, such as the case of the crossbreeding programs of Menz sheep, where selection led to the detriment of the black color, causing a decrease in its frequency and, consequently, in the population productivity, since evidence dictates that parameters such as birth weight, growth rate, yearling weight, and breeding values are higher in the black genotype than in the white genotype [83]. In this sense, the effects mentioned before in local goats have been partially studied, so it is, therefore, of the utmost importance to deepen the study of the environmental and genetic factors involved in production.

## 6. Economic Importance of the Coat Color

The industry’s preference for specific skin colors and fiber characteristics, such as diameter, length, and color, has guided the selection of particular coat colors [16]. For ex-ample, black goats from West Africa have the highest economic values [63]. However, the opposite situation occurs in Ethiopia, since black goats are the least-preferred animals, being offered at discounts of up to 15% compared to other colors [83].

White coats tend to have a higher economic value, as the white fleece can be dyed to create coats of any color. This situation has been changing due to the growing interest in natural colors, which are related to a higher preoccupation of consumers with the natural environment and the preference for natural products [16].

Some efforts have allowed the development of expression profiles for white, black, and brown colors in goats; this provides the opportunity to intervene at a molecular level to control the color in fibers that reach high economic values [84]. In the same way, there is evidence indicating that white color in the Angora breed is the most common and dominant as a result of a strong artificial selection for white fibers [49]; this is why directional regulation of the color of coats in fiber-producing animals is an important characteristic that has a direct impact on the price [36] and, in consequence, on the profitability of the activity.

## 7. Coat Color as a Selection Criterion in Breeding Programs

Most of the efforts to improve the production yield in livestock farms have focused on quantitative features, including aspects that are known to affect the profitability of the companies, because the aim is to maximize economic profit. Nevertheless, this has led to leaving aside the observable qualitative features that include color and appearance, which impact yield and productive life, meaning they constitute an essential part of the selection criteria in the production systems, mainly in those which are low-scaled [85,86]. In this sense, the objectives of genetic improvement are often multifaceted, and different approaches must have considerations beyond productivity. Therefore, since the integration of the concept of breed, the application of management practices has led to the creation of well-defined breeds, standardizing phenotypic characteristics primarily associated with phaneroptic and morphological aspects [87].

Some reports indicate that, in agro-pastoral production systems, such as the ones described in places from Africa and Mexico, the selection of bucks is made by appearance, with a predilection to spotted or mottled animals and flat colored [88,89,90] (Figure 1). In addition, several factors of social nature have a principal role in the definition of selection criteria. For example, in Ethiopia, goats are bred with a variety of colors for some specific religious rituals [91]. As such, the distribution of color variations within local populations is closely related to the preference of stakeholders, market demand (meat and skin), adaptability, litter size, mortality, and milk production potential, among others [2].

Berhanu et al. [64] and Adedeji et al. [57] state that producers employ indirect selection criteria based on the pigmentation of the coat, as this is one of the most significant characteristics in the selection of bucks compared with the does. In addition to the body size, the presence of horns and relative behavioral history (without using production records) are used, thus making it difficult to develop specific selection criteria. Hence, developing indirect selection schemes acquires relevance for these production systems [57,92]. In this respect, Lee et al. [93] found that coat color directly affects Holstein cattle’s longevity. However, in goats, this information is minimal. Some efforts have been made in northern Mexico [94] that have shed light on a possible effect of coat and skin color on local goats’ longevity and phenotypic differentiation. Nevertheless, these data need to be corroborated and studied in more detail, and, in the case that results are verified, this would be advantageous as an indirect selection criterion, since greater ease of recognition of desirable animals and greater longevity are directly related to a longer productive life [90,94]. The animal’s color is one of the most important means of identification between and within herds [64].

It is possible that including morphologic characteristics related to adaptation, such as coat color, is a relevant element in genetic improvement programs, which could ensure the survival of the animals [95]. Another benefit is that small producers can obtain an in-direct and economical method to improve the yield of local populations; otherwise, this is an impossible task for a classic genetic improvement program because of the stake-holders’ low educational level and lack of training [6,86,96]. 

**Table 2 biology-12-00929-t002:** Color diversity and productive response in goats.

Breed	Coat Color	Productive Trait	Reference
West African Dwarf	Light color	Increase milk production.	[72]
Salem	Black	Better meat quality.	[80]
Saanen	White	Increase milk production.	[59]
Alpine	Dark	Decline milk production under thermal stress.	[59]
Black Bengal	Black	Better adult weight, daily milk yield, lactation period, age at puberty, service per conception, litter size, abortion rate, kid mortality.	[69,74]
Tunisian local	Black	Fertility rate (87%), prolificacy rate (130%).	[82]
Indigenous	White, brown	Big size, tall body frame, long legs.	[83]
Local	Dark	Live weight, zoometric measures.	[97]

## 8. Conclusions

There has been a crucial technologic advance towards understanding the genetic expression of the coat color in principle, as given by genomic and last-generation molecular tools. However, these tools have not reached most producers of local breeds in developing countries since most of these studies have focused on characterizing populations in developed countries. Nevertheless, small producers use coat color as a primary criterion for selecting their animals. Notwithstanding, it is necessary to carry out studies associated with this feature that provide certainty to producers to guide their choices in order to make genetic improvements. Future efforts must be focused on evaluating these genotypes’ productivity, considering the application of genomic and bio-technologic tools to design valid genetic improvement schemes which are cost-effective and of massive applicability, in order to help increase productivity and preserve the genetic diversity of these local populations.

## Figures and Tables

**Figure 1 biology-12-00929-f001:**
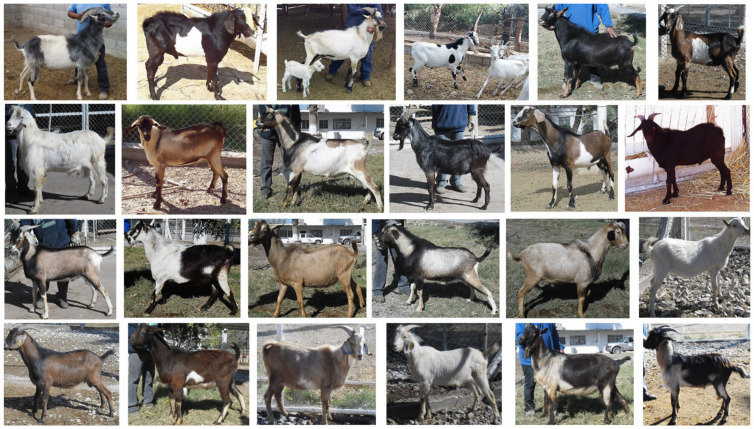
Diversity of color patterns in local bucks (*Capra hircus*) from Northern Mexico.

## Data Availability

Not applicable.

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
