# Peer review of "Coat Color in Local Goats: Influence on Environmental Adaptation and Productivity, and Use as a Selection Criterion"

_biology, 2023, doi:10.3390/biology12070929_

Round 1

Reviewer 1 Report

This review paper's main objective is to provide a comprehensive examination of the fragmented information available on the importance of coat color in local goats and its relationship with productivity and other traits. The paper appears to be logically organized and covers a comprehensive range of relevant topics on the significance of coat color in goats.References are extensively cited.

It is a nice review paper in my view though I have a couple of minor suggestions for the paper. None of my comment will sink the paper.

1. In Figure 1, please include the species names for the goats depicted. This additional detail will provide greater context and specificity for the reader.

2. Between Lines 120-146, I suggest the inclusion of a brief explanation on the techniques used in previous studies to identify pertinent genes. For instance, did the researchers utilize whole-genome data or other types of data? This added insight will offer readers a better understanding of the methodologies employed in these genomic analyses.

Author Response

The authors appreciate the time and observations made to our manuscript entitled “Coat Color in Local Goats: Influence on Environmental Adaptation, Productivity and as a Selection Criterion”. Likewise, we allow ourselves to respond to the observations made.

Throughout the manuscript, all observations including added references are indicated in red.

In this document, the comments made by the reviewer are included and those we consider requests for improvement to the manuscript are marked in red and their response is immediately marked in yellow.

Reviewer 2 Report

Manuscript ID: biology-2428187

Title: Coat Color in Local Goats: Influence on Environmental Adaptation, Productivity and as a Selection Criterion

This manuscript reviewed the importance of coat color in local goats and its relationship with other traits. Results could help small producers to have selection criteria to select elite individual. However, this manuscript only have a little content on genetic mechanism underlying coat color variation (L168-181). I have several comments to improve this manuscript.

Major concerns:

1.In “2. Expression of coat color and its role in racial conformation ” section, have a little content on genetic mechanism underlying coat color variation (L168-181), please review more literature on  genetic mechanism.

2.In “5. Milk and meat production”, please add a table to list the relationship between coat color and production traits.

3.Figure 1, the content of figure 1 is NOT sufficient, please add more coat color variation in goat. A reference “Kalds et al. Genetics Selection Evolution (2022) 54:61  https://doi.org/10.1186/s12711-022-00753-3” might be useful to improve Figure1.

Minor concern:

Table 1, It is better to list the same breed together, for example, in table 1, row 8 “Lianoning Cashmere ” can be put in row 3.

Author Response

(The authors gave the same response as above.)
